

# Identification and analysis of novel recessive alleles for *Tan1* and *Tan2* in sorghum

Lixia Zhang[1], Chunyu Wang[1], Miao Yu[2], Ling Cong[1], Zhenxing Zhu[1], Bingru Chen[2] and Xiaochun Lu[1]

[1] Sorghum Research Institute, Liaoning Academy of Agricultural Sciences, Shenyang, Shenhe, China
[2] Institute of Crop Germplasm Resources, Jilin Academy of Agricultural Sciences, Gongzhuling, Kemaoxi Street, China

Corresponding authors
Bingru Chen,
chenbingru1979@163.com
Xiaochun Lu,
luxiaochun2000@126.com

## ABSTRACT

**Background:** The identification and analysis of allelic variation are important bases for crop diversity research, trait domestication and molecular marker development. Grain tannin content is a very important quality trait in sorghum. Higher tannin levels in sorghum grains are usually required when breeding varieties resistant to bird damage or those used for brewing liquor. Non-tannin-producing or low-tannin-producing sorghum accessions are commonly used for food and forage. *Tan1* and *Tan2*, two important cloned genes, regulate tannin biosynthesis in sorghum, and mutations in one or two genes will result in low or no tannin content in sorghum grains. Even if sorghum accessions contain dominant *Tan1* and *Tan2*, the tannin contents are distributed from low to high, and there must be other new alleles of the known regulatory genes or new unknown genes contributing to tannin production.
**Methods:** The two parents 8R306 and 8R191 did not have any known recessive alleles for *Tan1* and *Tan2*, and it was speculated that they probably both had dominant *Tan1* and *Tan2* genotypes. However, the phenotypes of two parents were different; 8R306 had tannins and 8R191 had non-tannins in the grains, so these two parents were constructed as a RIL population. Bulked segregant analysis (BSA) was used to determine other new alleles of *Tan1* and *Tan2* or new Tannin locus. *Tan1* and *Tan2* full-length sequences and tannin contents were detected in wild sorghum resources, landraces and cultivars.
**Results:** We identified two novel recessive *tan1-d* and *tan1-e* alleles and four recessive *Tan2* alleles, named as *tan2-d*, *tan2-e*, *tan2-f*, and *tan2-g*. These recessive alleles led to loss of function of Tan1 and Tan2, and low or no tannin content in sorghum grains. The loss-of-function alleles of *tan1-e* and *tan2-e* were only found in Chinese landraces, and other alleles were found in landraces and cultivars grown all around the world. *tan1-a* and *tan1-b* were detected in foreign landraces, Chinese cultivars and foreign cultivars, but not in Chinese landraces.
**Conclusion:** These results implied that *Tan1* and *Tan2* recessive alleles had different geographically distribution in the worldwide, but not all recessive alleles had been used in breeding. The discovery of these new alleles provided new germplasm resources for breeding sorghum cultivars for food and feed, and for developing molecular markers for low-tannin or non-tannin cultivar-assisted breeding in sorghum.

## INTRODUCTION

Sorghum (*S. bicolor* (L.) Moench) is the fifth largest food crop in the world and widely used for producing food, feed, brewed beverages and biofuel (*Dahlberg, 2019*; *Zhao et al., 2019*). Tannins (also known as condensed tannins or proanthocyanidins) are important for the perception of quality in sorghum, and the tannin content determines the use of sorghum grains. Tannins are widespread in fruits, nuts, vegetables and some cereals (*He et al., 2008*). During crop domestication and evolutionary processes, tannin production is removed from major cereal crops (such as rice, wheat, and maize) but is retained in finger millet, barley, and sorghum (*Zhu, 2019*). Tannins have diverse biological and biochemical functions. Higher contents of anthocyanins and tannin compounds in sorghum grains can prevent bird attacks. In China, high-tannin-producing sorghum grains are particularly important in liquor production, accounting for 80% of China's total sorghum production. The well-known Moutaijiu, Langjiu, Luzhoulaojiao, Wuliangye and several other famous liquors are fermented by using high-tannin-producing sorghum grains as main feedstock (*Zhang et al., 2022*). Tannins not only inhibit the growth of miscellaneous bacteria but also produce syringic acid and syringaldehyde essential for the unique flavor during the brewing process. Moreover, distinct tannin content can brew different aroma, taste, and flavor liquors. However, high-tannin-producing grains and plants have negative impacts on nutritional value, such as decreasing protein digestibility and feed efficiency in humans and animals (*Choi & Kim, 2020*; *Chung et al., 1998*). Meanwhile, low tannin content can promote human health because of high antioxidant capacity and ability to fight obesity through reduced digestion (*Cos et al., 2004*; *Habyarimana et al., 2019*). Therefore, non-tannin-producing or low-tannin-producing sorghum cultivars are used in food and feeding production. Thus, it is very important to breed elite sorghum varieties with suitable tannin contents to meet different needs for food, feed, and liquor brewing industry. Tannin-producing and non-tannin-producing (or low-tannin-producing) sorghum cultivars are widely grown worldwide for their different applications and economic values. Evaluating the data on the presence of tannins in 11,557 cultivated sorghum accessions in Africa, approximately 55% are of the non-tannin-producing type and 45% are of the tannin-producing type (*Wu et al., 2019*). The coexistence of tannin-producing and non-tannin-producing (or low-tannin-producing) sorghum suggests that the elimination of this compound from sorghum grains during domestication is incomplete, exemplifying strong artificial selection against tannins in breeding and production.

Tannins (proanthocyanidins) and anthocyanins are major flavonoid end-products from a well-conserved family of aromatic molecules that have several biological functions in plant development and defense (*Gutierrez, Avila & Torres, 2020*; *Huang et al., 2019*; *Xie et al., 2019*; *Xie & Xu, 2019*). Tannins are derived from a branch of the flavonoid pathway, as well documented in *Arabidopsis*. AtTT2/AtTT8/AtTTG1 forms an MBW complex (MYB-bHLH-WD40) to regulate tannin synthesis (*Baudry et al., 2004*; *Li et al., 2020*; *Schaart et al., 2013*; *Wang et al., 2017*). Using genetic linkage mapping, *Tannin1* (*Tan1*,

*Sobic.004G280800*) and *Tannin2* (*Tan2, Sobic.002G076600*) have been cloned in sorghum (*Wu et al., 2019*, *2012*). Tan1 encodes a WD-40 repeat protein, and Tan2 encodes a bHLH domain protein, both have a regulatory function similar to that of *Arabidopsis* AtTTG1 and AtTT8. Three loss-of-function alleles each for *Tan1* and *Tan2* were identified in sorghum, including *tan1-a, tan1-b, tan1-c, tan2-a, tan2-b*, and *tan2-c*. Low or no tannins in sorghum grains can result from recessive alleles at one or both of *Tan1* and *Tan2* loci. A genome-wide association study (GWAS) was used to detect other tannin-related loci to explain natural variation in grain tannin content and pigmentation. Three highly significant association peaks spanning were observed, including 1.16–1.23 Mb (Chr1), 8.075–8.45 Mb (Chr2) and 57.9 Mb (Chr3) (*Morris et al., 2013*), were different from *Tan1* and *Tan2*. The reported data showed that other genes controlling tannin production may exist.

To develop practical molecular markers for tannin breeding, more *Tan1* and *Tan2* alleles need to be detected. We used wild sorghum resources, as well as landraces and cultivars, to comprehensively identify the alleles of *Tan1* and *Tan2*. We identified two novel recessive *Tan1* alleles and four recessive *Tan2* alleles by map-based cloning and sequencing *Tan1* and *Tan2* coding sequences. These new alleles will provide a solid foundation to study the evolution of *Tan1* and *Tan2* and their artificial selection in cultivar breeding and provide genetic resources for breeding non-tannin-producing or low-tannin-producing sorghum cultivars.

## MATERIALS AND METHODS

### Plant materials

Sorghum accessions include wild sorghum resources, landraces and cultivars from all over the world, were collected from the Sorghum Research Institute, Liaoning Academy of Agricultural Sciences, China. Plants were grown at the experimental site of Liaoning Academy of Agricultural Sciences (Shenyang (41.8°N, 123.4°E)). Each sorghum accession was planted in a single 3-m-long plot with 0.6-m row spacing. Leaf tissue was collected, frozen in liquid nitrogen and stored at −80 °C for DNA extracting. Grains were harvested to determine the tannin contents.

### DNA extraction

Leaves from each sorghum accession were sampled for genomic DNA extraction by the cetyltrimethylammonium bromide (CTAB) method as previously described with minor modifications (*Allen et al., 2006*). Add 2.5 volume of ethanol and incubate at room temperature for 10 min to precipitate DNAs. The mixtures were centrifuged under 13,000 g at room temperature for 10 min to pellet the DNAs. The DNA pellets were directly air-dried at room temperature for 5 min, and then dissolved in nuclease-free water.

### PCR, DNA sequencing, and sequence analysis

To genotype *Tan1* and *Tan2* alleles in different sorghum accessions, primers were designed (Table S1). The PCR products were sequenced by Beijing Tsingke Biological Technology Co., Ltd. (Beijing, China). The DNAMAN program (version 5.2.2) was used for sequence

alignment and translation of nucleotides into amino acids. *Tan1-1F/ Tan1-1R* primers were designed for detecting *tan1-a* and *tan1-b* (Table S1). To develop the CAPS (cleaved amplified polymorphic sequence) marker to detect *tan1-c* allele, *Tan1-2F/ Tan1-2R* combined with *Dde* I were designed for *tan1-c* (Table S1). PCR products were digested with *Dde* I and analyzed by 8% polyacrylamide gel electrophoresis. Because *tan1-c* lost a *Dde* I restriction enzyme site as a result of A-to-T transversion at position 1054 in the coding sequence, PCR amplification with *Tan1-2F/ Tan1-2R* resulted in a 164 bp product (G deletion at position 1057); whereas dominant *Tan1* contained a single *Dde* I site in the corresponding PCR product (165 bp) and was cut into 109 bp and 56 bp fragments. Primers of *Tan2-1F/Tan2-1R* to *Tan2-6F/ Tan2-6R* were designed for detecting *Tan2* alleles (Table S1).

## Determination of tannin content by reagent test kit

Tannin was determined according to the Tannin Microplate Assay Kit (Cohesion Biosciences, CAK1060). Five grams of grains were crushed into powder in a grinder. Tissue samples (0.1 g) were homogenized with 1 ml distilled water, placed in a water bath at 80 °C for 30 min, and centrifuged at 8,000 g at 4 °C for 10 min. The supernatant was placed into a new centrifuge tube for detection. 10 μl sample supernatant, 160 μl distilled water and 20 μl reaction buffer were mixed and incubated for 5 min at room temperature. Then, 10 μl dye reagent was mixed for 10 min, and the absorbance was measured and recorded at 650 nm to calculate the tannin content. The tannin contents were scored as low (≤0.5%), medium (0.5%> tannin content <1.0%) and high (≥1.0%). In this study, medium tannin content accessions were not exhibited.

## Mapping population

8R191 was a Chinese landrace without tannins. 8R306 was a landrace from Africa and had tannins in the grains. The RIL population "8R306 × 8R191" was obtained by advancing random individual $F_2$ plants to the $F_6$ generation by single-seed descent with 557 lines.

## Chlorox bleach test

Chlorox bleach test was performed previously described with minor modifications (*Dykes, 2019*). A total of 100 sorghum grains was placed into a 100 ml beaker, and 15 ml 6% NaClO was added to fully immerse sorghum grains. The beaker was left to sit for 20 min at room temperature, and the contents were swirled in the beaker every 5 min. The reaction solution, was discarded then it was rinsed with distilled water 2–3 times, and poured on filter paper to remove excess water. All bleach tests were repeated three times.

The presence or absence of tannins in sorghum grains was evaluated based on grain color after dyeing. Sorghum grains were divided into three types: Type I grains were completely black and had tannins, Type II grains were lighter brown black or had small black spots, and Type III grains were white or lightly colored and had no tannins. In 557 RILs, Type I had 269 lines, Type II had 179 lines and Type III had 109 lines. For the accuracy of phenotypic identification, Type I and Type III lines were used to map the new *Tannin*

locus. After dyeing, one grain was selected from one line to use for germinating and the seedling was sampled to extract the genomic DNAs.

## Mapping and identification of the candidate gene

Genomic DNAs were extracted from 8R191, 8R306 and RIL population plants using the CTAB method. BSA (Bulked Segregant Analysis) was used to determine the new *Tannin* locus. Equal amounts of genomic DNAs from 50 tannin (Type I, 50/269) and 50 non-tannin (Type III, 50/109) plants were pooled to construct the tannin and non-tannin bulks, respectively. Whole-genome resequencing was performed using Illumina HiSeq2000 platform by Beijing PlantTech Biotechnology Co., Ltd (Beijing, China). The depths of two bulks and two parental lines were about 50× and 10×, respectively. The reads were aligned to the *Sorghum bicolor* v3.1.1 reference genome (https://phytozome-next.jgi.doe.gov/info/Sbicolor_v3_1_1) using Burrows-Wheeler Aligner (BWA) software bwa-0.7.10 (*Li & Durbin, 2009*), GATK toolkit used to detect and filter SNPs (*McKenna et al., 2010*). The SNPs were employed as input in the R package "QTLseqr" version 0.7.5.2 for subsequent analysis (*Takagi et al., 2013*). To perform the SNP-INDEX analysis, the window size was set at 1 Mb. The significance of ΔSNP-index was determined at a 99% confidence interval and at least 10 SNPs within the window size. Ten InDel markers within the region were used for fine mapping. Information on molecular markers for fine mapping was provided in Table S1.

# RESULTS

## Relationships between *Tan1* and *Tan2* genotypes and tannin contents

The presence of tannins in sorghum grains is regulated by a pair of genes (*Tan1* and *Tan2*), and both genes have three recessive alleles (*Wu et al., 2019*, *2012*). Twenty accessions were used to determine the relationship between *Tan1* and *Tan2* genotypes and tannin contents in sorghum grains. Primers of *Tan1-1F/ Tan1-1R*, *Tan1-2F/ Tan1-2R* and *Tan2-1F/Tan2-1R* to *Tan2-6F/ Tan2-6R* were used to detect *Tan1* and *Tan2* alleles in different sorghum accessions. Tannin presence was scored by the Tannin Microplate Assay Kit for sorghum grains. As shown in Table 1, homozygous recessive genotypes at one or both genes can cause low-tannin phenotypes, and two wild sorghum resources had high tannin contents because they carry dominant alleles, which was consistent with the reported data (*Wu et al., 2019*). However, six accessions carrying *Tan1* and *Tan2* dominant alleles had low tannin contents in sorghum grains, indicating that there may be variation in unknown genes contributing to tannin production or new alleles of the known regulatory genes (Table 1).

## A novel recessive allele of *Tan1* in sorghum landrace

To identify new tannin genes or new alleles, a recombinant inbred line (RIL) population was built from 8R191 and 8R306. Sequencing and digestion were used to determine the *Tan1* and *Tan2* alleles. Amplifying and sequencing by *Tan1-1F/ Tan1-1R* indicated that 8R191 and 8R306 didn't have *tan1-a* and *tan1-b* (Table S1). A CAPS marker was designed to detect the *tan1-c* allele. 165 bp PCR products with *Tan1-2F/2R* primers were digested by

**Table 1 Genotype and phenotype of 20 accessions.**

| Accessions | *Tan1* | *Tan2* | Phenotype | Origin | Germplasm type |
|---|---|---|---|---|---|
| 8R156 | *tan1-a* | *Tan2* | low-tannin | India | landrace |
| BTx623 | *tan1-b* | *Tan2* | low-tannin | United States | cultivar |
| Tx2752 | *tan1-c* | *Tan2* | low-tannin | United States | cultivar |
| 8R111 | *Tan1* | *tan2-a* | low-tannin | Senegal | landrace |
| 8R035 | *Tan1* | *tan2-a* | low-tannin | Mali | landrace |
| RTx430 | *tan1-a* | *tan2-a* | low-tannin | United States | cultivar |
| 8R374 | *Tan1* | *Tan2* | low-tannin | China | landrace |
| 8R336 | *Tan1* | *Tan2* | low-tannin | China | landrace |
| JS255 | *Tan1* | *Tan2* | low-tannin | China | landrace |
| JS257 | *Tan1* | *Tan2* | low-tannin | China | landrace |
| JS266 | *Tan1* | *Tan2* | low-tannin | China | landrace |
| JS273 | *Tan1* | *Tan2* | low-tannin | China | landrace |
| 8R245 | *Tan1* | *Tan2* | high-tannin | China | landrace |
| 8R249 | *Tan1* | *Tan2* | high-tannin | China | landrace |
| 8R284 | *Tan1* | *Tan2* | high-tannin | China | landrace |
| 8R243 | *Tan1* | *Tan2* | high-tannin | China | landrace |
| 8R446 | *Tan1* | *Tan2* | high-tannin | China | landrace |
| 8R312 | *Tan1* | *Tan2* | high-tannin | China | landrace |
| SV1-5 | *Tan1* | *Tan2* | high-tannin | NA | wild |
| TU11 | *Tan1* | *Tan2* | high-tannin | NA | wild |

*Dde* I, 109 bp and 56 bp DNA fragments of dominant *Tan1*. Because of A-to-T transversion at position 1054 in the coding sequence of *tan1-c*, the PCR products of Tx2752, OK11 and Tx631 remained uncleaved (Table S1, Fig. 1) (*Wu et al., 2019*). The PCR product of 8R191 and 8R306 can cleave indicating that two parents did not have *tan1-c*. Six primers were used to detect *Tan2* genotypes in 8R191 and 8R306 (Table S1). Tannin presence was scored by the chlorox bleach test for 8R191 and 8R306 grains. 8R191 is a non-tannin landrace and 8R306 is a tannin landrace, both of them carrying *Tan1* and *Tan2* dominant alleles.

We performed BSA using the 8R191/8R306 RIL population lines. Equal amounts of genomic DNAs from 50 tannin and 50 non-tannin plants were pooled to construct the tannin and non-tannin bulks, respectively. The 8R191(non-tannin parent), 8R306(tannin parent), and non-tannin, tannin bulks were subjected to Illumina high-throughput sequencing, from which, 70.99, 72.90, 290.71 and 280.82 million paired-end reads were produced, representing 14×, 14×, 54×, and 53× genome coverage, respectively (Table S2). Among them, 98.23%, 97.88%, 94.35% and 95.91% reads could be mapped to the *Sorghum bicolor* v3.1.1 reference genome, respectively, indicating good quality of the sequencing data (Table S2). Using the BSA-Seq method, we obtained only one region spanning 6.60 Mb on Chr4 between 57,400,000 to 64,000,000 was strongly associated with the tannin phenotype (Fig. 2A). Within this region, we developed 10 available InDel markers for fine

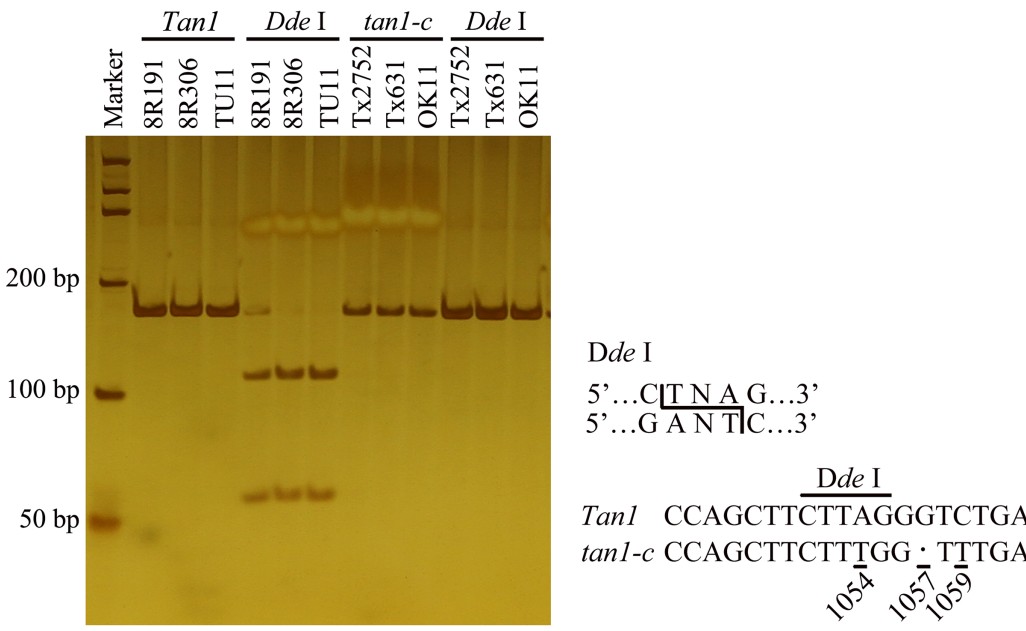

**Figure 1 Development of molecular marker for *Tan1* and *tan1-c* in sorghum.** The marker is from Wuhan Servicebio Technology Co., Ltd (GN100bp DNA Ladder I, G3365-01). A-to-T transversion at position 1054 in the coding sequence of *tan1-c* results in loss of a *Dde* I restriction site, but that is present in *Tan1*. The 164 bp PCR product (G deletion at position 1057) from *tan1-c* is uncleaved, but the 165 bp product from *Tan1* is cleaved into 109 bp and 56 bp fragments by *Dde* I. *Tan1*: 8R191, 8R306 and TU11 (wild sorghum); *tan1-c*: Tx2752, Tx631 and OK11 (*Wu et al., 2019*).

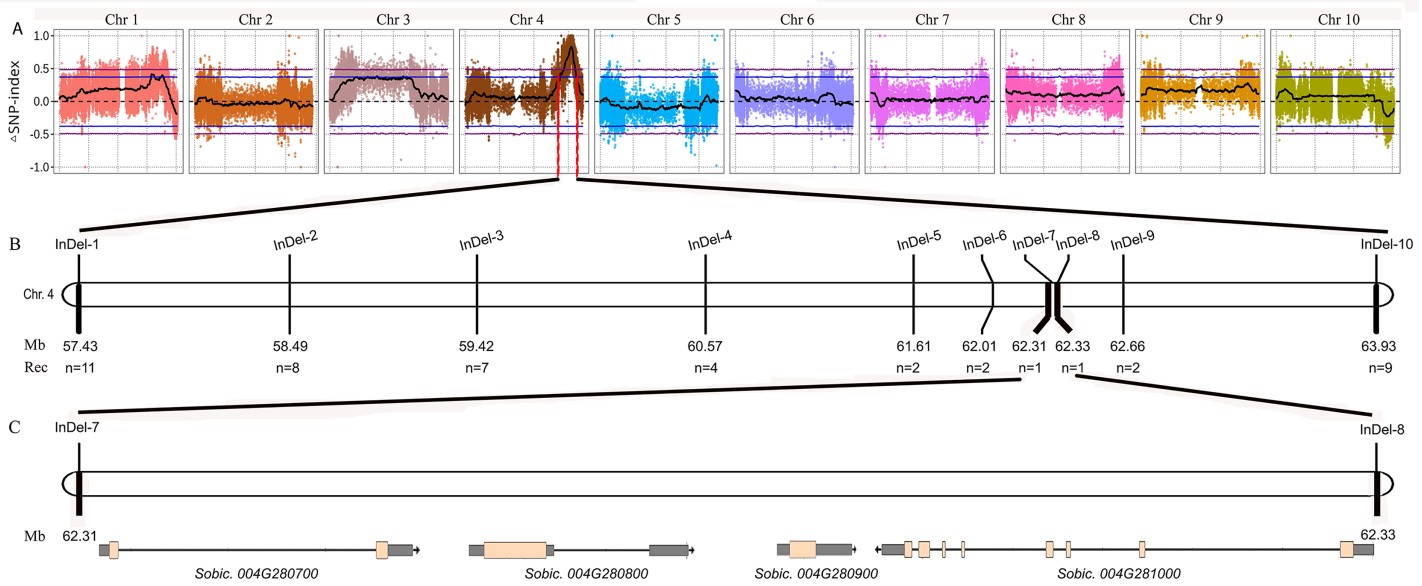

**Figure 2 Fine mapping of the *tan1-d*.** (A) ΔSNP index plot. ΔSNP index = SNP index (Tannin) − SNP index (Non-tannin). The purple dashed line represents the threshold (0.49) of ΔSNP index. The area above the purple dashed line is the rough mapping interval of *tan1-d* on Chr4. (B) Distribution of InDel markers in the rough mapping interval. The fine mapping interval is narrowed between InDel-7 and InDel-8. (C) Sobic.*004G280700*, Sobic.*004G280800* (*Tan1*), Sobic.*004G280900* and Sobic.*004G281000* are found in the fine mapping interval.

mapping (Fig. 2B). Using a RIL population of 378 plants (Type I: 269 lines and Type III: 109 lines), the new *Tannin* locus was finally narrowed down to a 25.8 kb region defined by markers *InDel-7* and *InDel-8*. We identified four candidate genes in this region, including *Sobic.004G280700*, *Sobic.004G280800*, *Sobic.004G280900* and *Sobic.004G281000* (Fig. 2C, Table S3). *Sobic.004G280800* is *Tan1*, suggesting that *Sobic.004G280800* is likely to be the causal gene. Primers (*Tan1-F*/*Tan1-R*; Table S1) were used to detect the sequence polymorphisms in *Tan1* between 8R191 and 8R306. 8R306 had dominant *Tan1*, however, 8R191 had deletion and substitution in coding region of *Tan1*, and named as *tan1-d*. Therefore, *Tan1* has another new genotype that affects its function.

## Identification and distribution of *Tan1* and *Tan2* other new alleles

To identify more different *Tan1* and *Tan2* alleles, we collected 396 sorghum accessions, including wild sorghum resources, landraces and cultivars (Table S4). DNA was extracted using leaves by CTAB method. The PCR primers *Tan1-F*/*Tan1-R* were designed for detecting *Tan1* alleles. Primers from *Tan2-1F*/*Tan2-1R* to *Tan2-6F*/ *Tan2-6R* were designed for detecting *Tan2* alleles (Table S1). Tannin content was determined according to the Tannin Microplate Assay Kit. According to the coding sequence variation and low-tannin content, the new variation types were determined for *Tan1* and *Tan2*.

Another novel *Tan1* recessive allele was found, named as *tan1-e*. Among 396 sorghum accessions, 10 wild sorghum resources and 200 high-tannin-producing accessions carried dominant *Tan1*, and 89 low-tannin-producing accessions carried *tan1-a*, *tan1-b*, *tan1-c*, *tan1-d*, and *tan1-e* alleles. A total of 97 accessions carried the *Tan1* allele, but had low tannin contents, accounting for 24.5% (97/396) of the total sample size (Table 2, Table S4). Maybe there was allelic variation of *Tan2* in low-tannin-producing accessions with dominant *Tan1*. Then seventy-two accessions carrying the *Tan1* allele were used to detect different *Tan2* alleles, including 38 low-tannin-producing accessions, 24 high-tannin-producing accessions and 10 wild sorghum resources. Ten wild sorghum resources and 24 high-tannin-producing accessions had dominant *Tan1* and *Tan2* (Table 3, Table S5). Four different recessive alleles of *Tan2* were identified, named as *tan2-d*, *tan2-e*, *tan2-f*, and *tan2-g* (Table 3, Table S5). 24 sorghum accessions carried the dominant *Tan1* and *Tan2* alleles, but had low tannin contents, accounting for 63.2% of 38 low-tannin sorghum accessions (Table 3). Our analysis indicates that there may be existed other unknown loci involved in tannin production.

More importantly, *Tan1* and *Tan2* recessive alleles had obvious regional distribution characteristics. Novel *tan1-d*, *tan2-d*, *tan2-f* and *tan2-g* alleles were distributed worldwide, including Ghana, France, Mexico, India, China and so on, but *tan1-e* and *tan2-e* were only found in Chinese landraces (Tables S4 and S5). The results implied that some *Tan1* and *Tan2* recessive alleles had different geographical distributions.

## Functional variation of the newly identified *Tan1* and *Tan2* recessive alleles

*Tan1* and *Tan2* are conserved regulatory factors in the plant tannin synthesis pathway and have higher nucleotide similarity within major cereal crops other than rice, wheat and

**Table 2 Genotype and phenotype of 396 accessions.**

| Phenotype | Tan1 | Accession number |
|---|---|---|
| Low-tannin | Tan1 | 97 |
| | tan1-a | 46 |
| | tan1-b | 18 |
| | tan1-c | 14 |
| | tan1-d | 9 |
| | tan1-e | 2 |
| High-tannin | Tan1 | 200 |
| High-tannin (wild) | Tan1 | 10 |
| Total | | 396 |

**Table 3 Genotype and phenotype of 72 accessions.**

| Phenotype | Genotype | Accession number |
|---|---|---|
| | Tan1/Tan2 | 24 |
| | Tan1/tan2-a | 3 |
| Low-tannin | Tan1/tan2-d | 1 |
| | Tan1/tan2-e | 6 |
| | Tan1/tan2-f | 1 |
| | Tan1/tan2-g | 3 |
| High-tannin | Tan1/Tan2 | 24 |
| High-tannin (wild) | Tan1/Tan2 | 10 |
| Total | | 72 |

maize, which do not produce tannins in their grains. Tan1 encodes a WD-40 repeat protein and has four WD-40 repeat domains. Deletion, substitution and insertion mutations in *tan1-a*, *tan1-b*, and *tan1-c* have caused frame shifts and premature stop codons, leading to disruption of the highly conserved region of WD-40 domain and C-terminus and resulting in the absence or low level of tannins in sorghum grains. Seven independent mutations of the *TTG1* gene reveal that the truncation of the C-terminal region and WD-40 domain produced nonfunctional alleles in *Arabidopsis*, indicating that the C-terminal region and WD-40 domain are vital for the structure and function of WD-40 protein (*Wu et al., 2012*). In *tan1-d*, A-to-T transversion at position 1054, GT deletion at positions 1057 and 1058, and C-to-T transition at position 1059 in the coding sequence affected TGA (at positions 1060, 1061 and 1062) stop codon frameshift and led to nonfunctional protein. Because of mutation and deletion, *tan1-d* had 1086 bp coding sequence and 361 aa protein sequence. Sequence variation of *tan1-d* was similar to *tan1-c* and four WD-40 domains presented, but the C-terminal sequence had changed greatly in tan1-d. Compared to dominant *Tan1*, *tan1-e* had a 10-bp deletion (CGACATACGT) in the coding sequence between positions 771 and 780. The 10-bp deletion caused a

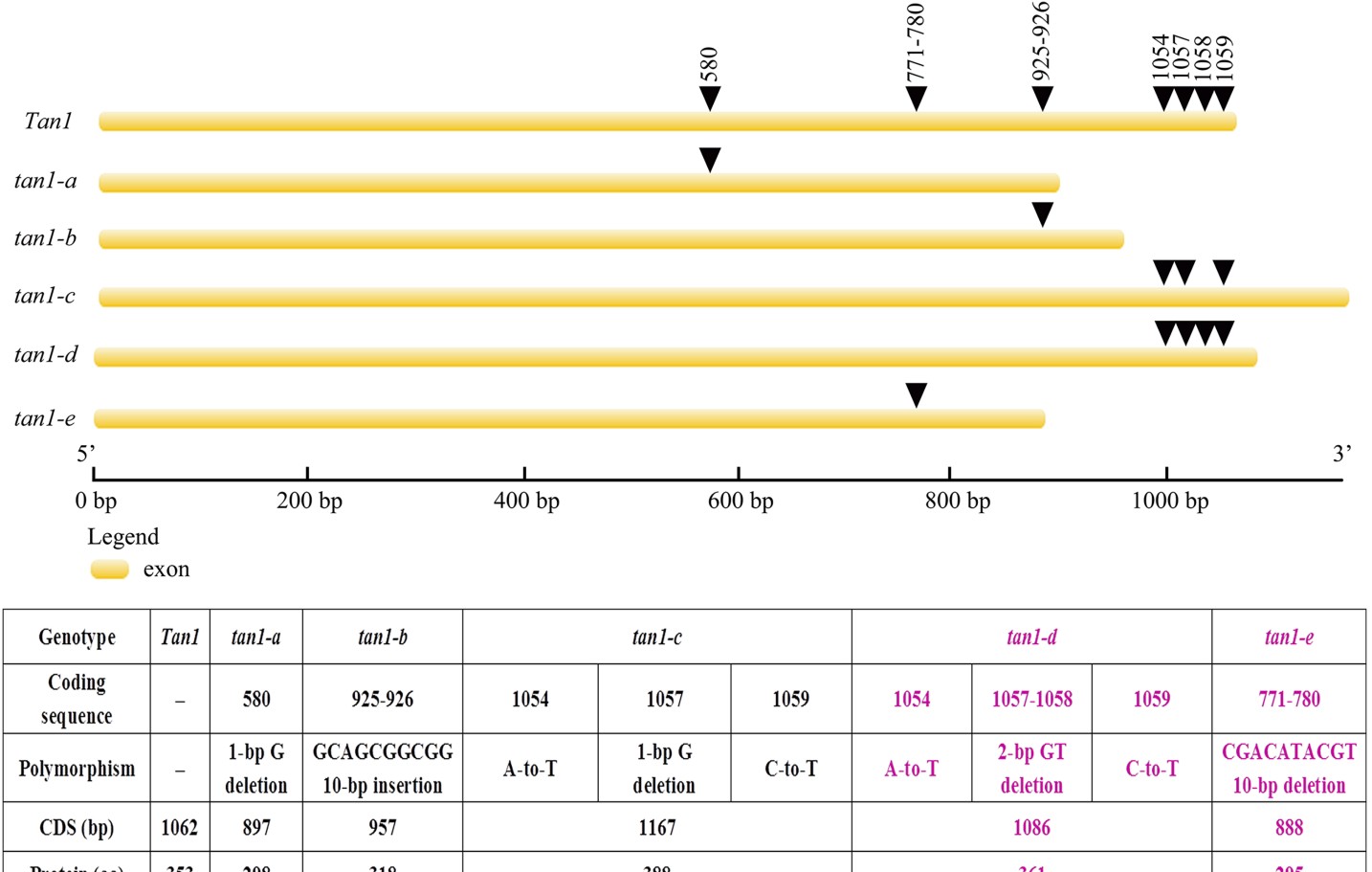

| Genotype | Tan1 | tan1-a | tan1-b | tan1-c | | | tan1-d | | | tan1-e |
|---|---|---|---|---|---|---|---|---|---|---|
| Coding sequence | – | 580 | 925-926 | 1054 | 1057 | 1059 | 1054 | 1057-1058 | 1059 | 771-780 |
| Polymorphism | – | 1-bp G deletion | GCAGCGGCGG 10-bp insertion | A-to-T | 1-bp G deletion | C-to-T | A-to-T | 2-bp GT deletion | C-to-T | CGACATACGT 10-bp deletion |
| CDS (bp) | 1062 | 897 | 957 | 1167 | | | 1086 | | | 888 |
| Protein (aa) | 353 | 298 | 318 | 388 | | | 361 | | | 295 |

**Figure 3 Gene structures and mutation sites for *Tan1* alleles.** *tan1-a*, *tan1-b*, and *tan1-c* were reported previously (*Wu et al., 2012, 2019*) and *tan1-d* and *tan1-e* were identified in this work.             

frameshift mutation and resulted in a truncated protein with a length of only 295 aa. The fourth WD-40 domain and C-terminal region were dramatically changed in tan1-e (Figs. 3 and 4, Fig. S1).

*Tan2* encodes a bHLH transcription factor with 10 exons and nine introns in BTx623. *tan2-a* has a 5-bp (CTCCC) insertion in the 8th exon, *tan2-b* has a 7-bp (CCACAGA) insertion in the 8th exon and *tan2-c* has a 95-bp deletion removing the entire 8th intron (Fig. 5 and Fig. S2). Three mutations lead to frame shift, disrupt the bHLH domain and result in non-tannin-producing or low-tannin-producing phenotype (*Wu et al., 2019*). *tan2-d*, with the causal polymorphism of a 1-bp C deletion at position 563 in the coding region, led to a truncated protein with a length of only 210 aa. Because of the C-to-T transition at position 1366 (**C**AG to **T**AG) in the coding sequence, *tan2-e* resulted in premature termination and had a 455 aa protein. *tan2-f* contained a frameshift mutation and an early termination site because of an 8-bp (AGCTGATC) insertion between positions 1375 and 1376 in the coding region, resulting in a 462 aa protein sequence. tan2-d, tan2-e and tan2-f disrupted the bHLH domain structure and lost function. *tan2-g*, a null

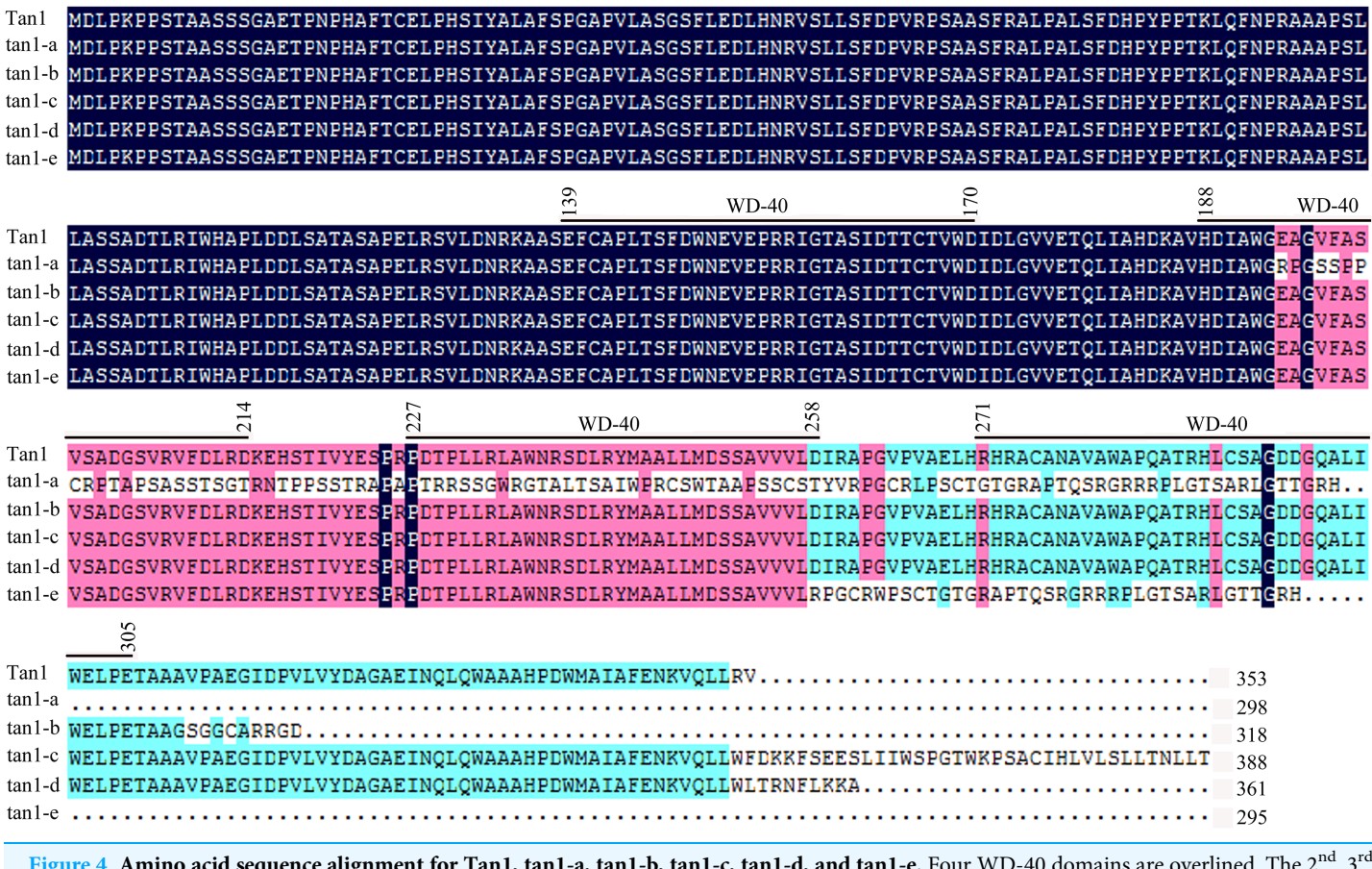

**Figure 4 Amino acid sequence alignment for Tan1, tan1-a, tan1-b, tan1-c, tan1-d, and tan1-e.** Four WD-40 domains are overlined. The 2[nd], 3[rd] and 4[th] WD-40 domains are missed in tan1-a and 4[th] WD-40 domain is missed in tan1-e. Although four WD-40 domains present in tan1-b, tan1-c, and tan1-d, C-terminal sequences have dramatically changed.

allele, had multiple substitutions and insertions from position 1579 to 1607 in the coding region and didn't disrupt the bHLH domain structure (Figs. 5 and 6, Fig. S2).

### *Tan1* and *Tan2* alleles utilization in breeding programs

By investigating the distribution of *Tan1* and *Tan2* alleles in sorghum cultivars, we can determine which alleles have been used in breeding. 87 cultivars (from China and foreign countries), as well as 34 sterile lines and 43 restorer lines, were used to detect the different alleles of *Tan1* and *Tan2* (Tables S4 and S6). For *Tan1*, only *tan1-a*, *tan1-b* and *tan1-c* alleles were detected in Chinese cultivars, *tan1-d* and *tan1-e* may not have been inherited in low-tannin-producing cultivars (Tables S4 and S6). For *Tan2*, *tan2-a*, *tan2-b* and *tan2-c* were detected in cultivars in the reported data (*Wu et al., 2019*). In our study, *tan2-f* and *tan2-g* alleles were detected in cultivars, and *tan2-d* and *tan2-e* alleles were not (Tables S4 and S5). More importantly, *tan1-e* and *tan2-e* were only detected in Chinese landraces (Table S5). The results showed that only some *Tan1* and *Tan2* alleles were applied in breeding, leading to a decrease in the diversity of breeding resources.

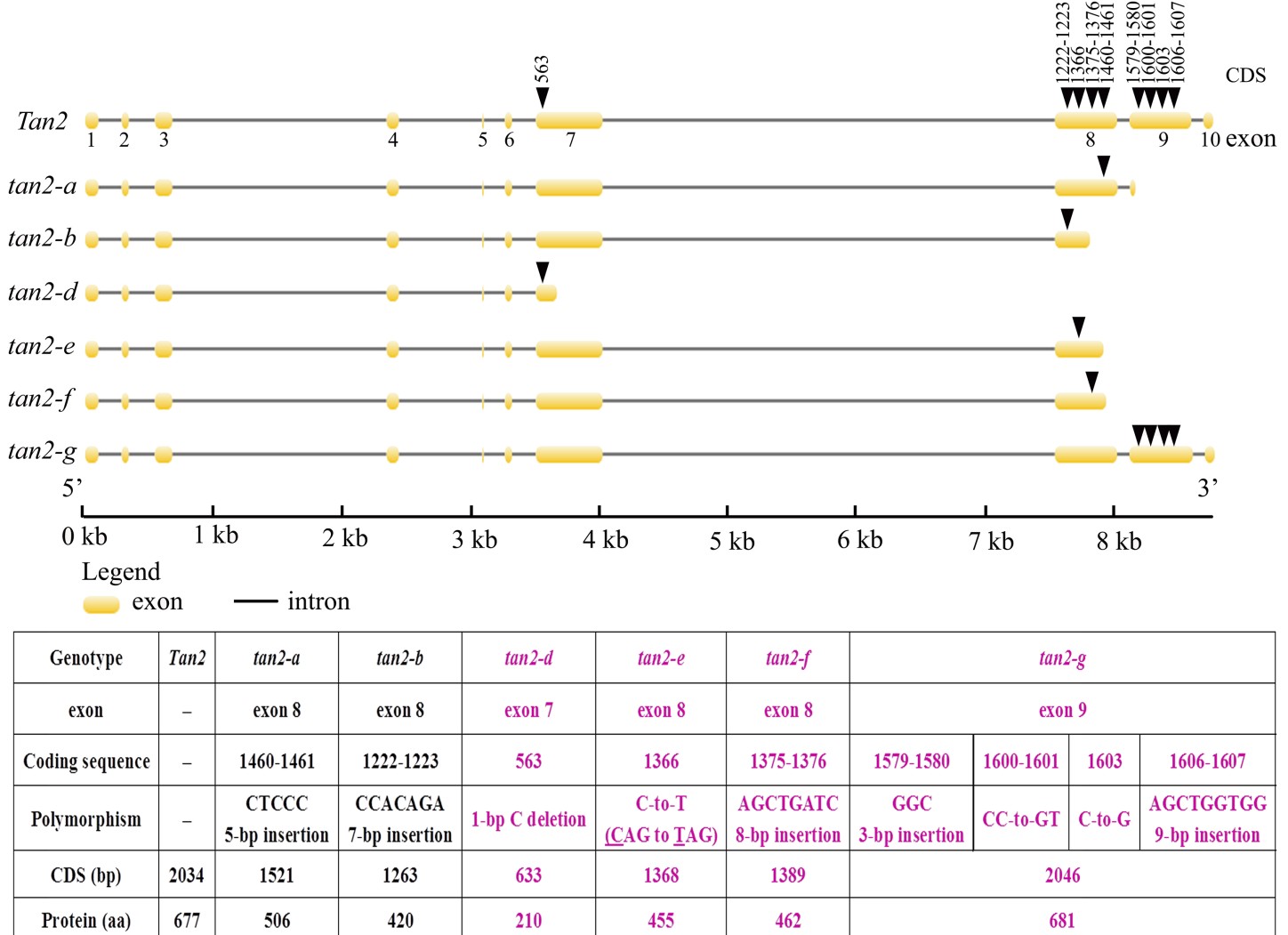

**Figure 5** **Gene structures and mutation sites for *Tan2* alleles.** *tan2-a*, *tan2-b*, and *tan2-c* were reported previously (*Wu et al., 2019*) and *tan2-d*, *tan2-e*, *tan2-f*, and *tan2-g* were identified in this work.

# DISCUSSION

Wild sorghum resources generally show higher tannin contents than domesticated accessions due to selection during domestication (*Dykes & Rooney, 2007*). The apparent nutrient absorption and protein digestion issues were reduced by feeding sorghum grains with high tannin content. Breeders mainly rely on grain color to determine the contents of tannins in grains (*Rhodes et al., 2014*). Sorghum accessions with pigmented testa usually contain condensed tannins. The use of grain color as a proxy for tannin concentration is complicated by the need for varietal information, including pigmented testa and endosperm appearance, which are correlated with tannin levels (*Dykes, 2019*; *Oliveira et al., 2017*). In fact, grain color is not a reliable indicator of sorghum tannin contents. Using marker-assisted breeding can simplify and expedite breeding for determining the

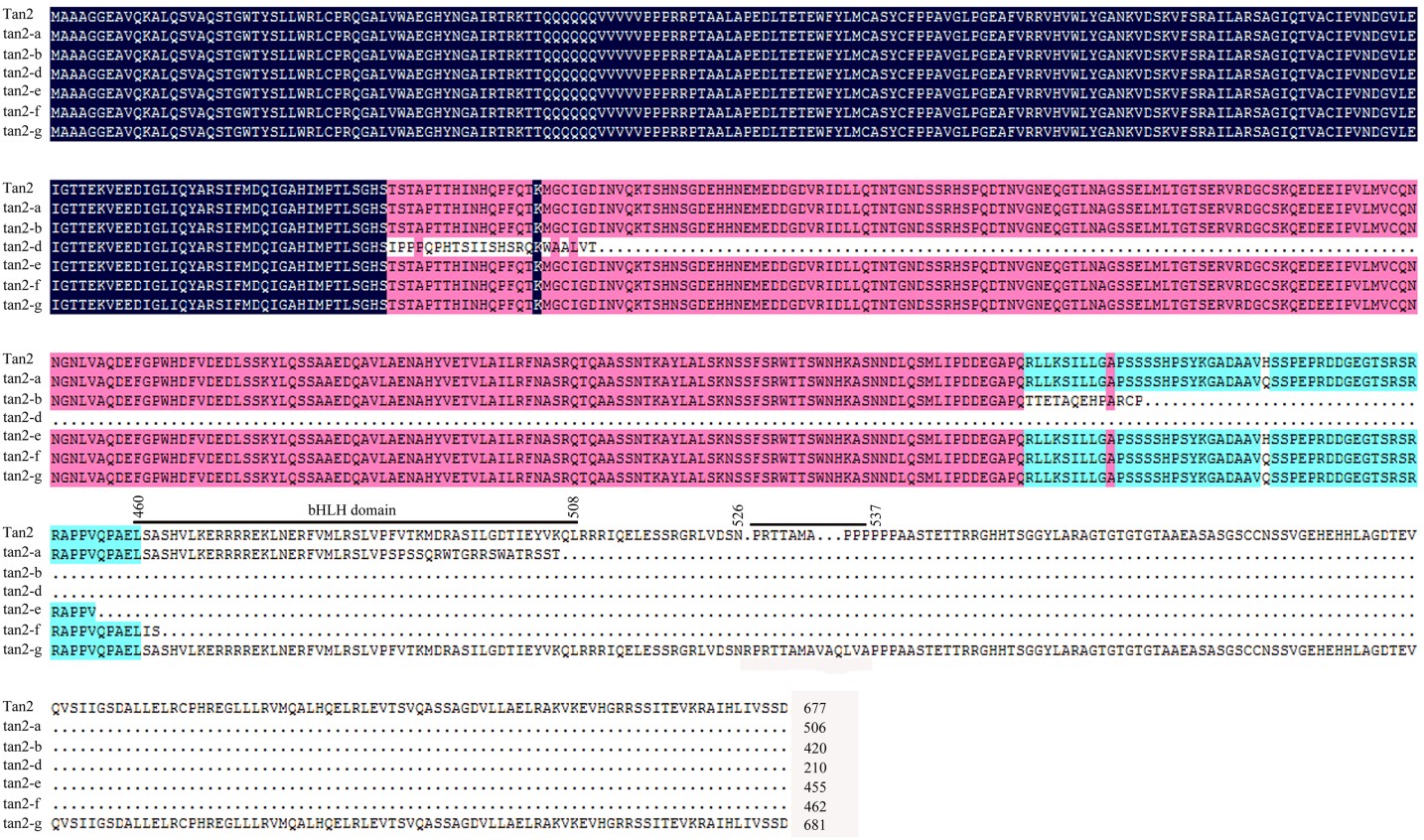

**Figure 6 Amino acid sequence alignment for Tan2, tan2-a, tan2-b, tan2-d, tan2-e, tan2-f, and tan2-g.** The conserved bHLH domain, which is missed in tan2-a, tan2-b, tan2-d, tan2-e, and tan2-f, is overlined. Although bHLH domain present in tan2-g, amino acid sequences between 526 and 537 have greatly changed.

tannin content. Identifying tannin-related genes and alleles is very important for molecular selection and breeding.

AtTT2, AtTT8 and AtTTG1 form an MBW complex to regulate tannin synthesis (*Baudry et al., 2004*; *Ha et al., 2018*; *Schaart et al., 2013*). Nonfunctional AtTTG1 and AtTT8 proteins impact MBW complex function, which inhibits the expression of *DFR*, *LAR* and *ANR* and hinders tannin synthesis (*Shan et al., 2019*; *Sun et al., 2022*; *Wei et al., 2019*). *Tan1* (homologous gene-*AtTTG1*) and *Tan2* (homologous gene-*AtTT8*) are involved in regulating the tannin synthesis pathway in sorghum, and three recessive alleles each for *Tan1* and *Tan2* have been reported (*Wu et al., 2019*, *2012*). In our study, two novel recessive alleles for *Tan1* and four novel recessive alleles for *Tan2* were identified, including *tan1-d*, *tan1-e*, *tan2-d*, *tan2-e*, *tan2-f*, and *tan2-g* (Figs. 3 and 5, Table S7). Because of insertion or deletion in the coding regions of five recessive *Tan1* alleles and seven recessive *Tan2* alleles, their corresponding Tan1 and Tan2 proteins are nonfunctional and show variable inhibition of tannin accumulation in sorghum grains. These alleles will be useful for marker-assisted breeding for the improvement of low-tannin-producing or non-tannin-producing sorghum cultivars.

The tannin contents of 186 out of 396 accessions were under 0.5%. These accessions were widely distributed in China, India, Africa and other countries (Tables S4 and S5). These low-tannin-producing accessions contain recessive *Tan1* and *Tan2* alleles or dominant *Tan1* and *Tan2* alleles. According to the reported data, sorghum accessions with dominant *Tan1* and *Tan2* alleles should contain high tannins, but in our study the tannin contents of some sorghum accessions with dominant *Tan1* and *Tan2* alleles were still low, indicating that there maybe unknown genes or new alleles of the known regulatory genes in the tannin synthesis pathway. Furthermore, the identified *Tan1* and *Tan2* alleles have certain characteristics of regional distribution; for example, *tan1-e* and *tan2-e* are only distributed in China (Tables S4 and S5).

There are many different characteristics among Chinese sorghum accessions, African sorghum accessions and Indian sorghum accessions. Heterosis in Chinese sorghum accessions is also different from that in African sorghum accessions and Indian sorghum accessions. However, these data cannot be regarded as evidence of a Chinese or foreign origin for sorghum but can indicate that Chinese sorghum accessions have high diversity and a strong evolutionary history. In our study, *tan1-a* and *tan1-b* were not detected in Chinese landraces (Tables S4 and S6). In Chinese sterile and restored lines, 16 materials contained *tan1-a* allele and 11 materials contained *tan1-b* allele, respectively (Table S6). Meanwhile, eight Chinese cultivars had *tan1-a* allele and five Chinese cultivars had *tan1-b* allele, indicating that *tan1-a* and *tan1-b* alleles may come from foreign accessions (Table S4).

The *Tan1* genotypes were detected in 145 accessions of foreign sorghum (landraces and cultivars) with low tannin contents, however, there was no *tan1-e* allele in low-tannin-producing foreign accessions (Table S4). As shown in our data, *tan1-e* was only detected in two Chinese landraces. The two *tan1-e* landraces are from Jilin Province and Shanxi Province in China. The genetic background of these two accessions is quite different as there is 700 km between the two provinces. These results suggest that different alleles of *Tan1* may have different geographic distributions and selective advantages in sorghum breeding. *Tan2* and six recessive alleles (*tan2-a*, *tan2-b*, *tan2-c*, *tan2-d*, *tan2-f*, and *tan2-g*) were found in the United States, West Africa, Western Europe, North America, India, China and other parts of the world. However, the recessive *tan2-e* allele was only found in Chinese landraces (Table 3, Table S5). Therefore, *Tan1* and *Tan2* may serve as important clues to study the origin and evolutionary history of Chinese sorghum and foreign sorghum.

## CONCLUSIONS

In our study, two new allelic variants of *Tan1* and four new allelic variants of *Tan2* were identified. Up to now, five recessive alleles of *Tan1* and seven recessive alleles of *Tan2* alleles were found, indicating that *Tan1* and *Tan2* had abundant allelic variants. This was because of loss-of-function recessive alleles in *Tan1* and *Tan2*, which lead to low or no tannin content in sorghum grain. Only *tan1-e* and *tan2-e* were found, *tan1-a* and *tan1-b* were not found in Chinese landraces, and other alleles were found in landraces or cultivars worldwide. Some *Tan1* and *Tan2* alleles have not been used in breeding.

### Funding

This research was supported by the China Agricultural Research System (CARS-06-14.5-A3), China Postdoctoral Science Foundation (2021M693847), the Liaoning Province Special Program for Germplasm Innovation and Store Grain in Technology (2023JH1/10200009), and the Jilin Province Science and Technology Development Program (20210202116NC). The funders had no role in study design, data collection and analysis, decision to publish, or preparation of the manuscript.

### Grant Disclosures

The following grant information was disclosed by the authors:
China Agricultural Research System: CARS-06-14.5-A3.
China Postdoctoral Science Foundation: 2021M693847.
Liaoning Province Special Program for Germplasm Innovation and Store Grain in Technology: 2023JH1/10200009.
Jilin Province Science and Technology Development Program: 20210202116NC.

### Competing Interests

The authors declare that they have no competing interests.

### Author Contributions

- Lixia Zhang conceived and designed the experiments, performed the experiments, analyzed the data, prepared figures and/or tables, authored or reviewed drafts of the article, and approved the final draft.
- Chunyu Wang performed the experiments, prepared figures and/or tables, and approved the final draft.
- Miao Yu performed the experiments, prepared figures and/or tables, and approved the final draft.
- Ling Cong analyzed the data, prepared figures and/or tables, authored or reviewed drafts of the article, and approved the final draft.
- Zhenxing Zhu analyzed the data, authored or reviewed drafts of the article, and approved the final draft.
- Bingru Chen analyzed the data, authored or reviewed drafts of the article, and approved the final draft.
- Xiaochun Lu conceived and designed the experiments, prepared figures and/or tables, authored or reviewed drafts of the article, and approved the final draft.

### Data Availability

The sequences are available at the NGDC GenBase: C_AA012100.1 (tan1-d), C_AA012101.1 (tan1-e), C_AA012102.1 (tan2-d), C_AA012103.1 (tan2-e), C_AA012104.1 (tan2-f) and C_AA012105.1 (tan2-g).

https://ngdc.cncb.ac.cn/genbase/review/16038a94c884

## Supplemental Information

Supplemental information for this article can be found online at http://dx.doi.org/10.7717/peerj.17438#supplemental-information.

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
