# Peer review of "Identification and analysis of novel recessive alleles for Tan1 and Tan2 in sorghum"

_PeerJ, doi:10.7717/peerj.17438_

## Round 0.1 · original submission · Major Revisions

Dear Authors

Based on the referees' comments, the manuscript is not acceptable for publication in its present form. However, if the authors can suitably address the reviewers' comments, I invite them to revise and resubmit their manuscript.

Best Regards

Reviewer 1 ·

Basic reporting

Major concerns:
1. For the figures, I suggest the authors should give the very clear gene and protein structural diagrams for Tan1, Tan2, and all of their mutated forms.
2. Figure 3 showed the comparison of Tan1, tan1-d and tan1-e. I am very confused that what is the meaning of figure 4? Besides, what is the differences between figure 5 and figure 6? I am also surprised that the figure legends are same in figure 5 and 6.
3. I am confused of the logical structure. I wonder that why not show the BSA analysis first? A good logic is: detection of the Tannin1, followed by new alleles of tan1, new tan2 alleles in the other paragraph. Two or three figures are enough: BSA, Tan1 alleles, Tan2 alleles.
4. In figure 1, The authors are suggested to show a road map of CAPS marker developing. It is better to put the CAPS marker-related information at the end of figures.
5. This manuscript shows the poor writing. There are numerous grammar mistakes and misleading saying in the text. These need to be checked.

Minor concerns:
1. Sorghums?
2. In the last two sentences of background and Line 165-167, what do you mean the other alleles on the base of Tan1Tan2 in some accessions?
3. Line 84-86, this sentence may be modified due to the confused saying.
4. Line 129-131, what is the generation of the RIL?
5. Line 158, I did not see the relationships between Tan1/Tan2 genotypes and tannin contents. The authors may perform a visual association analysis.
6. Line 214-215, what is the evidence of Tan1 and Tan2 may have been selected artificially after independent evolution in ecotype areas.
7. Figure 3, the entire predicted protein structure of tan1-c should be given.

Experimental design

no comment

Validity of the findings

Zhang et al identified the novel alleles of known Tan1 and Tan2 in sorghum. It is a simple result. I am concerned about this manuscript that needs to be largely revised before it is considered to be published.

Reviewer 2 ·

Basic reporting

Overall, the study is interesting and has some new findings that may be important for breeding when tannin production is of interest. There are some places where the wording is ambiguous or confusing:

Lines 166-167 ...indicating that there may be variation in unknown genes contributing to tannin production or new alleles of the known regulatory genes.

Lines 190-195. Define type I and type III RIL lines. Also, I would not call this a "new" tannin locus as the map position is right on top of the Tan1 locus.

Line 198. Therefore, Tan1 has additional genotypes that affect its function

Line 200-201--Indicate briefly how you identified the new alleles

Line 204-214--of all the lines that were examined, does allelic variation (new alleles or known alleles) explain tannin deficiency in all lines? This is important as it will indicate if other unknown loci may be involved.

Lines 214-215--I think it only implies that different starting material (alleles) were present in China as opposed to other countries where many of the alleles are not geographically distributed.

Line 239-241 the URL needs to be separated by a space from the rest of the sentence

Lines 269-272--I think the authors have this reversed (nutrient and protein digestion issues reduced by feeding grains with low tannin content)

Lines 293-296--What proportion of the low tannin lines have dominant Tan1 and Tan2?

Lines 303-312--seems to be redundant with Results section, condense or remove.

Line 326-318--confusing. The claim is made initially that the genetic backgrounds due to the geographic distance but also suggests that they may have shared a common ancestor of the tan1-e allele...

Experimental design

Experimental design is okay. The study abruptly changes focus after the BSA analysis failed to reveal new loci involved in tannin production. Most questions above (basic reporting) are clarification

Validity of the findings

Validity seems to be okay. Some conclusions need to be better justified or better articulated and these are noted in the basic reporting section

·

Basic reporting

The manuscript is well written but still needs some minor changes

Experimental design

No comments

Validity of the findings

No comments

Additional comments

I reviewed the paper titled "Identification and analysis of novel recessive alleles for Tan1 and Tan2 in sorghum". The authors used wild sorghum, as well as landraces and cultivars, to comprehensively identify the alleles of tan1 and tan2. They identified two novel recessive tan1 alleles and four recessive tan2 alleles by map-based cloning and sequencing Tan1 and Tan2 coding regions.
-Comments and Suggestions for Authors
Abstract
- Please find more corrections as track changes in the manuscript pdf file.
Introduction
-The introduction section is comprehensive and well written.
-Please find more corrections as track changes in the manuscript pdf file.
Materials and methods
-Please find more corrections as track changes in the manuscript pdf file.
Results
-The results section is well written.
- Please find more corrections as track changes in the manuscript pdf file.
Discussion
-The discussion section is well written.
- Please find more corrections as track changes in the manuscript pdf file.
Conclusion
-The conclusion section is well written.
References
Please unify the style according to the journal instructions

·

Basic reporting

The article demonstrates clear and unambiguous writing, employing professional English throughout. The language used is appropriate for a scientific publication. The article provides adequate literature references, citing previous studies that are relevant to the topic of tannin synthesis regulation in sorghum. The references support the background information and provide a context for the research.
In terms of professional article structure, the work follows a conventional format with sections such as Introduction, Materials and Methods, Results, Discussion, and Conclusions. However, it is important to note that the level of detail in the Materials and Methods section could be improved by providing more specific information about the experimental procedures, sample sizes, and data analysis methods.
I suggest that the authors better to provide brief summaries of the referenced studies (e.g., Baudry et al. 2004; Ha et al. 2018; Schaart et al. 2013) to provide more context and clarify their relevance to the current research. This will help readers understand how these studies connect to the findings presented in the text.

Experimental design

The research question investigating the regulation of tannin synthesis in sorghum is well defined and relevant, addressing a specific knowledge gap in the field. The text highlights the identification of novel recessive alleles for Tan1 and Tan2 in sorghum (e.g., tan1-d, tan1-e, tan2-d, tan2-e, tan2-f, and tan2-g). This demonstrates original primary research within the journal's scope and contributes to expanding our understanding of tannin synthesis regulation. However, while the research question is clearly stated, the text does not explicitly explain how this research fills the identified knowledge gap. Providing a brief statement on the significance and potential implications of the findings in advancing our knowledge of tannin synthesis regulation would strengthen the justification for the study.
The methods employed in the research are briefly described, mentioning the identification of recessive alleles and their impact on tannin accumulation. However, additional details such as sample sizes, controls, and statistical analyses are lacking. Including these details would allow readers to better evaluate the rigour and reproducibility of the study. I suggest providing more detailed information about the applied criteria and the methods used, such as sample sizes, controls, and specific statistical analyses employed. This will allow readers to assess the rigour and reproducibility of the research and enhance the credibility of the findings.

Validity of the findings

The text presents interesting findings related to the identification of novel recessive alleles for Tan1 and Tan2 in sorghum, indicating potential variations in tannin synthesis regulation. However, the impact and novelty of these findings are not explicitly assessed or discussed. It would be valuable to provide an interpretation of the findings in the context of existing knowledge and highlight their significance for the field of tannin synthesis research. I suggest to:
a) evaluate and discuss the impact and novelty of the findings in relation to existing knowledge.
b) highlight the significance of the identified novel recessive alleles for Tan1 and Tan2.
c) explain how they contribute to advancing our understanding of tannin synthesis regulation.
This will provide a more comprehensive interpretation of the findings and their broader implications.

Additional comments

Per section, point-by-point critics and improvement suggestion for the authors to consider:
1. Title
I suggest a different title: “Understanding the Genetic Basis of Tannin Regulation: Insights from Novel Recessive Alleles of Tan1 and Tan2 in Sorghum”.
2. Introduction
The introduction provides a comprehensive background on the importance of tannins in sorghum and the need for identifying new alleles of the Tan1 and Tan2 genes. It effectively highlights the significance of tannins in sorghum grains and their impact on various applications and nutritional aspects. However, there are a few areas where additional information or clarification could enhance the introduction:
- Provide a concise overview of how tannins contribute to bird resistance and brewing processes, as mentioned in the introduction regarding the need for higher tannin levels in sorghum grains.
- Expand on the potential health benefits of tannins, such as their high antioxidant capacity and impact on digestion, to enhance understanding of their role in human nutrition.
- Explain the significance of genome-wide association studies (GWAS) in identifying additional genes or loci controlling tannin content, as mentioned in relation to association peaks on chromosomes 1, 2, and 3.
- Clarify the rationale for using wild sorghum, landraces, and cultivars to study tan1 and tan2 alleles comprehensively, exploring their variation and implications.
- Enhance the focus of the research question in the introduction by explicitly addressing the regulation of tannin synthesis and the role of the Tan1 and Tan2 genes. To enhance the clarity of the scientific questions, the authors could consider revising the introduction by:
a) Authors should explicitly state the main question, e.g., "What are the novel recessive alleles for Tan1 and Tan2 in sorghum and their impact on tannin synthesis?"
b) More context is needed on the importance of tannin synthesis, e.g., crop yield, quality, and genetic improvement benefits.
c) A Clear explanation of the significance of identifying recessive alleles and filling the knowledge gap, e.g., previous focus on dominant alleles and functional implications is needed.
d) Introduction should be logically organized, providing background and rationale for the research on tannin synthesis in sorghum and the Tan1 and Tan2 genes.
3. Material and Methods
A. Material
The material section provides information on the plant materials used in the study, specifically sorghum accessions and the mapping population. While the section covers the basics, there are a few areas where additional details could be helpful:
- The sorghum accessions used in the study include wild sorghums, landraces, and cultivars collected from the Sorghum Institute, Liaoning Academy of Agricultural Sciences, China. Provide more information about the number or range of accessions used, the geographic origins of the accessions, and any specific characteristics or traits relevant to the study.
- The plants were grown at the experimental site of the Liaoning Academy of Agricultural Sciences. Providing additional details about the location, climate, or specific conditions of the experimental site could help in understanding the environmental context of the study.
- The section briefly mentions the mapping population obtained by single-seed descent from 8R306 and 8R109, resulting in 557 lines. It would be beneficial to provide more information about the rationale behind selecting these specific parental lines, any known characteristics or traits of these lines, and the significance of generating the mapping population for the study.
B. Methods
The methods section describes the procedures followed for DNA extraction, PCR, DNA sequencing, sequence analysis, and determination of tannin content. Here's a breakdown of the key elements and suggestions for improvement:
- Description of DNA extraction steps, but lacks CTAB method modifications.
- PCR, DNA sequencing, and analysis need more information on sequence analysis parameters and alignment algorithms.
- CAPS marker development describes the design and primer combination but lacks details on specific conditions or protocols.
- Bulk Segregant Analysis (BSA), the criteria for plant selection and phenotypic details needed.
- No information on sequencing depth or data analysis methods and details.
- More details on SNP analysis methodology are required, including software/tools used and significance threshold.
- InDel and SNP markers for fine mapping: Additional information needed on marker number, density, and genotyping procedures.
- Determination of tannin content use a Tannin Microplate Assay Kit for tannin content determination. The sample preparation, incubation, and absorbance measurement steps are described. It would be beneficial to provide more details on the specific procedure or protocol used with the assay kit and mention any controls or standards used for calibration. I would strongly advise against this test, as from what is written, the following concerns are raised:
a) Method lacks details on bleach concentration, temperature, immersion duration, and swirling frequency, introducing variability and hindering comparisons.
b) Reliance on visual inspection introduces bias; quantification using spectrophotometry or HPLC would enhance reliability.
c) Lack of quantitative data, as the method categorizes by colour, not tannin content; comprehensive analysis with chemical assays is needed for accurate comparisons.
d) Lack of controls and replicates undermines reliability and reproducibility.
e) No mention of sample size or statistical analysis necessary for valid conclusions.
f) Unclear tannin scoring: Insufficient criteria for low, medium, and high tannin content, and exclusion of medium tannin accessions not justified.
4. Results:
The results presented in the text have several limitations and areas that could be criticized:
- Small sample size (20 accessions) limits findings' representativeness, applicability, and generalizability.
- Absence of statistical analysis undermines the strength and significance of observed relationships.
- Incomplete genetic characterization that fails to consider other potential genetic factors influencing tannin contents, restricting comprehensive understanding.
- No replication or validation experiments were conducted to confirm observed associations, compromising reliability.
- Insufficient description of genotyping, sequencing, and tannin content determination methods, hindering assessment of accuracy and reliability.
- Insufficient analysis of Tan1 and Tan2 alleles' effectiveness or impact on tannin content improvement, limiting practical insights for crop improvement efforts.
5. Discussion
Overall, the discussion provides valuable insights into the genetic regulation of tannin synthesis in sorghum and the potential application of these findings in breeding programs. However, there are a few aspects that can be criticized:
- More specific information on allele identification methods, accession selection criteria, and statistical analysis should be provided for transparency and reproducibility.
- The study fails to discuss or consider potential confounding factors such as environmental conditions, agricultural practices, and genetic interactions, which may impact tannin contents.
- The functional consequences of the identified alleles on the MBW complex and tannin synthesis are not thoroughly explored, warranting a deeper investigation.
- A comparative analysis of the identified alleles in different sorghum populations globally would offer valuable insights into evolutionary patterns and genetic diversity.
- Further explanation and supporting evidence are needed to clarify the different selective advantages and evolutionary modes of Tan1 between Chinese and foreign sorghum populations.

---

## Round 0.2 · Minor Revisions

Dear Authors

The manuscript still needs a minor revision before being reconsidered for publication. The authors are invited to revise the paper, considering all the reviewers' suggestions. Please note that the requested changes are required for publication.

With Thanks

Reviewer 1 ·

Basic reporting

No Comment

Experimental design

No Comment

Validity of the findings

No Comment

Additional comments

This version is better in terms of text description, but the figures may need to be revised again. Particularly, it is difficult for me to quickly check the variations of new alleles using the output DNAMAN graphs. I have some minor concerns.

1. The figures 3 and 5 are still not visually accessible for readers to examine the functional variations. I suggest creating a gene structure diagram illustrating the exons and introns (the first version has no clear gene structure), with annotations in a bellowed table indicating the mutated sites, their types, and the predicted alterations in the protein. And highlight the important frameshift mutations in each allele.

2. It would be better to include the grain color morphology after bleaching for lines with different tan1/tan2 alleles.

3. In lines 138-139 of the methods section, the authors provided a quantitative phenotype evaluation for tannins, but I did not observe it in the results presented in any of the figures or tables.

4. Most sorghum lines are named by authors, and it would be beneficial if they could be connected with common PI numbers to increase the impact of the paper.

5. The word 'sorghum' is a Latin name of a plant species, with no plural form. If the authors want to indicate multiple instances, simply replace it with 'sorghum resources or accessions'.

6. Arabidopsis, gene ID, marker and primer names should be italic.

·

Basic reporting

no comment

Experimental design

no comment

Validity of the findings

no comment

Additional comments

The authors have made the changes I suggested in the last review. I recommend its publication in this journal.

---

## Round 0.3 · Minor Revisions

Dear Authors,

The manuscript still needs minor revision.

Line 28-29. It is unclear from this that one parent has tannins and the other does not. It is also unclear what the genotypes are: do both parents have Dominant Tan1 AND Tan2 alleles? Or one has Tan1 and one has Tan2? And do dominant alleles cause high or low tannins?

Line 30: but no new locus was identified. Reword

You need to include citations for GATK and BWA software.

Calculation of the SNP index for BSA mapping should be described in the methods.

Best Regards

---

## Round 0.4 · accepted · Accept

Dear Authors,

I am pleased to inform you that after the last round of revision, the manuscript has been improved, and it can be accepted for publication.

Congratulations on the acceptance of your manuscript and thank you for your interest in submitting your work to PeerJ.

With Thanks